# FOXP1 orchestrates neurogenesis in human cortical basal radial glial cells

**Seon Hye E. Park**[1,2]**, Ashwinikumar Kulkarni**[1,2]**, Genevieve Konopka** [1,2]*

1 Department of Neuroscience, UT Southwestern Medical Center, Dallas, Texas, United States of America,
2 Peter O'Donnell Jr. Brain Institute, UT Southwestern Medical Center, Dallas, Texas, United States of America

* Genevieve.Konopka@utsouthwestern.edu

**Data Availability Statement:** With the exception of the single-nuclei RNA-sequencing data, all other data are within the paper and its Supporting Information files. The single-nuclei RNA-sequencing data reported in this paper can be

## Abstract

During cortical development, human basal radial glial cells (bRGCs) are highly capable of sustained self-renewal and neurogenesis. Selective pressures on this cell type may have contributed to the evolution of the human neocortex, leading to an increase in cortical size. bRGCs have enriched expression for Forkhead Box P1 (FOXP1), a transcription factor implicated in neurodevelopmental disorders (NDDs) such as autism spectrum disorder. However, the cell type–specific roles of FOXP1 in bRGCs during cortical development remain unexplored. Here, we examine the requirement for FOXP1 gene expression regulation underlying the production of bRGCs using human brain organoids. We examine a developmental time point when FOXP1 expression is highest in the cortical progenitors, and the bRGCs, in particular, begin to actively produce neurons. With the loss of FOXP1, we show a reduction in the number of bRGCs, as well as reduced proliferation and differentiation of the remaining bRGCs, all of which lead to reduced numbers of excitatory cortical neurons over time. Using single-nuclei RNA sequencing and cell trajectory analysis, we uncover a role for FOXP1 in directing cortical progenitor proliferation and differentiation by regulating key signaling pathways related to neurogenesis and NDDs. Together, these results demonstrate that FOXP1 regulates human-specific features in early cortical development.

## Introduction

The neocortex consists of diverse cell types that are produced in a highly species-specific manner under strict spatiotemporal control throughout development. Compared to lissencephalic species, the gyrrencephalic human neocortex is endowed with an expanded outer subventricular zone (oSVZ) that is occupied by the basal progenitors such as the basal radial glial cells (bRGCs) [1–3]. The human bRGCs are known for their exceptional capacity for self-renewal, and neurogenesis. bRGCs are known to produce neurons through both indirect neurogenesis, a production of neurons via intermediate progenitors, and direct neurogenesis, a production of neurons directly from bRGCs. Both of these mechanisms contribute to the prolonged birth of excitatory neurons (ENs), which underlie human cortical expansion [4,5]. Gene expression

accessed at NCBI GEO (https://www.ncbi.nlm.nih.gov/geo/query/acc.cgi?acc=GSE195510). Code that was used to perform data pre-processing, clustering and differential gene expression analysis is available at GitHub repository (https://github.com/konopkalab/organoidseq). Transgenic iPSC cell lines generated in this study can be provided upon request using appropriate material transfer agreements with UT Southwestern Office for Technology Development (technologydevelopment@utsouthwestern.edu).

**Funding:** G.K. is a Jon Heighten Scholar in Autism Research and Townsend Distinguished Chair in Research on Autism Spectrum Disorders at UT Southwestern Medical Center. This work was supported by the Welch Foundation (I-1997-20190330), Simons Foundation (573689), NIMH (MH126481, MH102603, MH103517), NIDCD (DC014702), NINDS (NS115821), NHGRI (HG011641), and the James S. McDonnell Foundation 21st Century Science Initiative in Understanding Human Cognition – Scholar Award (220020467) to G.K. The funders had no role in study design, data collection and analysis, decision to publish, or preparation of the manuscript.

**Competing interests:** The authors have declared that no competing interests exist.

**Abbreviations:** AD-EGFP, adenovirus-expressing GFP; aRGC, apical radial glial cell; ASD, autism spectrum disorder; bRGC, basal radial glial cell; BSA, bovine serum albumin; ChIP-seq, chromatin immunoprecipitation sequencing; CP, cortical plate; CRISPRi, CRISPR inhibition; DEG, differentially expressed gene; DIV, days in vitro; DTL, dorsal telencephalic lineage; EN, excitatory neuron; ESC, embryonic stem cell; FMRP, Fragile X Mental Retardation Protein; FOXP1, Forkhead Box P1; GW, gestation week; ICC, immunocytochemistry; ID, intellectual disability; IPC, intermediate progenitor cel; KD, knockdown; KO, knockout; NDD, neurodevelopmental disorder; NDS, normal donkey serum; OR, odds ratio; oSVZ, outer subventricular zone; RIPA, radioimmunoprecipitation assay; RPCA, reciprocal PCA; RT-PCR, real-time PCR; rtTA, reverse tetracycline-controlled transactivator; scRNA-seq, single-cell RNA-sequencing; snRNA-seq, single-nuclei RNA-sequencing; TF, transcription factor; UMAP, uniform manifold approximation and projection; VZ, ventricular zone; WT, wild type.

in these cell types during early development is tightly regulated, as any abnormal changes at this stage may have irreversible consequences for brain development [6–8]. However, the genetic and molecular components involved in the formation of early cortical neural cells that give rise to the neocortex have not yet been fully defined.

A crucial regulator of the molecular mechanisms underlying cortical development is the transcription factor (TF) Forkhead Box P1 (FOXP1). FOXP1 has been linked to neurodevelopmental disorders (NDDs), such as autism spectrum disorder (ASD) and intellectual disability (ID) [9,10]. Previous studies have shown that either knockdown (KD) or knockout (KO) of FOXP1 leads to varying phenotypes reflecting abnormal neurogenesis in the cortex [11–13]. However, no study has examined the role of FOXP1 in basal progenitors in relation to cortical development [14]. A recent study detected the expression of FOXP1 in the ventricular zone (VZ) of the developing human cortex at gestation week (GW) 14, the last week of fetal development that FOXP1 expression remains high in the cortical progenitors [13]. The same study showed that FOXP1 was detected in as many as 70% of human bRGCs, whereas FOXP1-positive (FOXP1+) bRGCs were not observed in the mouse cortex at an equivalent developmental stage. Currently, there is very little known about the cell type–specific roles of FOXP1 in the cortex. Single-cell or single-nuclei RNA-sequencing (scRNA-seq and snRNA-seq, respectively) studies of other FOXP1+ cell types in the brain, such as the spiny projection neurons of the striatum, revealed that FOXP1 regulates distinct gene expression programs within each spiny projection neuron subtype [15]. These findings show that FOXP1 has cell type–specific contributions to the development of the striatum and suggest that a similar role may occur in the cortex as well.

In our study, we investigate several unanswered questions regarding the cell type–specific contribution of FOXP1 to human cortical development, specifically in regard to bRGCs. First, we wanted to determine whether FOXP1 regulates gene expression programs that are important for the development of bRGCs. Second, we wanted to examine if FOXP1 regulates the proliferation and differentiation of bRGCs. Third and last, we sought to identify changes in the expression of corticogenesis and NDD-relevant genes with loss of FOXP1 in a cell type–dependent manner. In this way, we could determine how a lack of FOXP1 leads to NDD-relevant features in the developing human cortex. To capture the FOXP1+ bRGCs that cannot be studied using mouse models, we utilized 3D human brain organoids in combination with snRNA-seq (**Fig 1A and 1B**). We manipulated FOXP1 expression using CRISPR/Cas9 and evaluated how the loss of FOXP1 affects the development of bRGCs and ENs, as well as differential gene expression in all cell types, within the brain organoids. Using this system, we found that the loss of FOXP1 negatively affects both bRGC production and the differentiation of ENs. snRNA-seq enabled us to determine the differentially regulated genes associated with neurogenesis and NDDs in bRGCs with loss of FOXP1.

## Results

### Generation of human brain organoids with loss of FOXP1

Disruption of the *FOXP1* locus results in a collection of cognitive and neurodevelopmental symptoms known as FOXP1 syndrome [10,16]. The individuals diagnosed with this disorder have loss of function mutations such as deletions, frameshifts, or de novo point mutations in 1 copy of *FOXP1* [16,17]. To study the essential function of FOXP1 in a manner that reflects human brain development, we applied 2 different CRISPR-mediated gene KO strategies at the stem cell stage of human brain organoids. The first strategy involved a complete gene deletion (KO-1) yielding no FOXP1 mRNA and protein. The second strategy employed a reporter knock-in into 1 allele and a double stranded break and frameshift in another allele (KO-2)

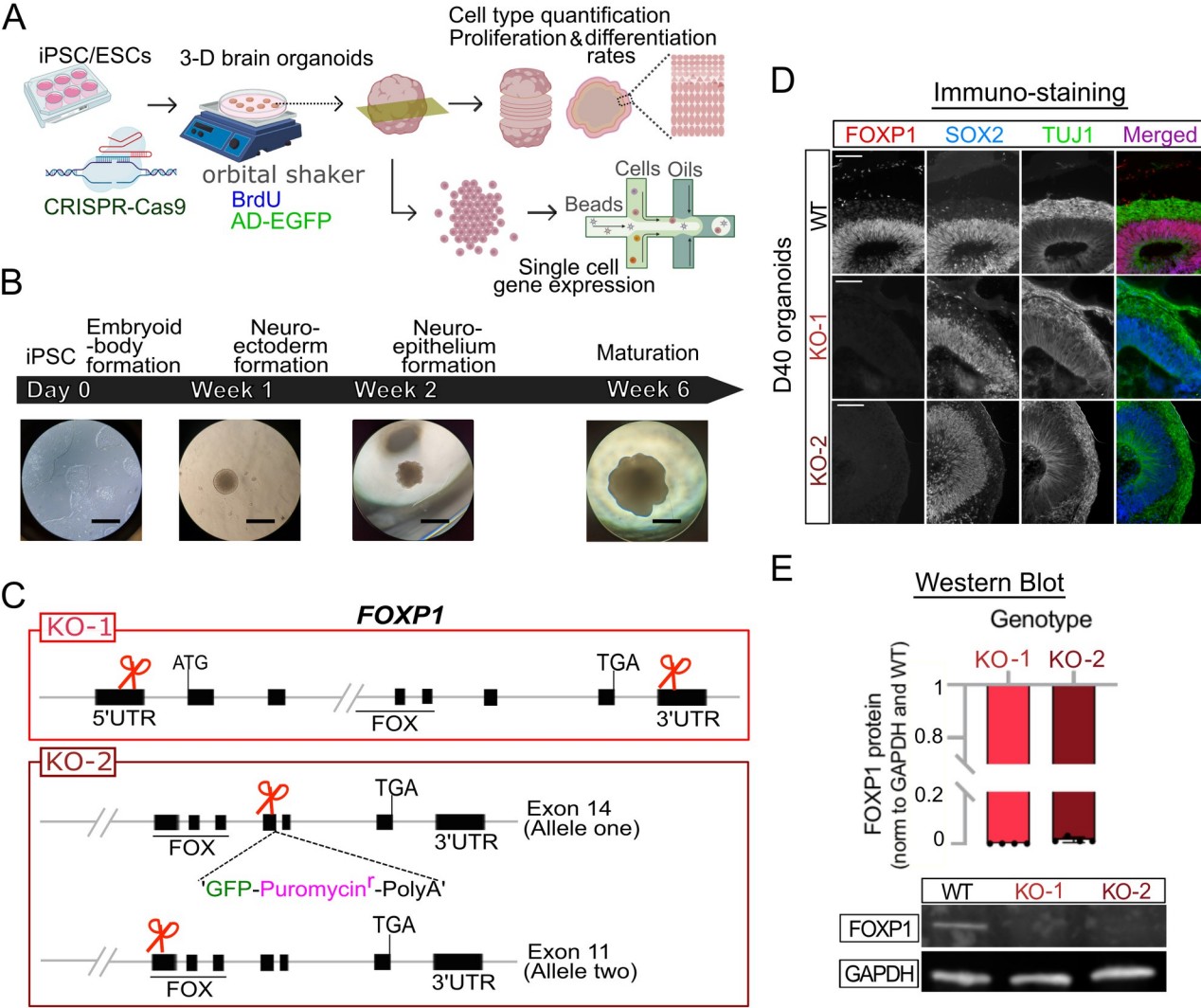

**Fig 1. Generation of FOXP1 KO cerebral organoids from stem cells. (A)** Schematic diagram showing the experimental design. Created with [BioRender.com](https://biorender.com). **(B)** Cerebral organoid protocol illustrating typical morphology observed at each time point during differentiation from embryoid body formation to maturation. Scale bar = 600 μM **(C)** The 2 CRISPR-Cas9 strategies used in generating FOXP1 KOs. **(D)** Representative immunostaining results showing FOXP1 expression in SOX2+ and TUJ1+ cells in WT and absence of FOXP1 expression in the KOs at D40 in vitro. Scale bar = 100 μM **(E)** Representative western blot results with quantification from $n = 4$. The numerical values that were used to generate the graph can be found in S1 Data. D40, day 40; FOXP1, Forkhead Box P1; KO, knockout; WT, wild type.

yielding a nonfunctional FOXP1 protein (**Fig 1C**). We then differentiated these edited stem cells into cerebral brain organoids (**Figs 1B and S1A–S1F and S1 Data**), which are a model system known to reliably recapitulate many characteristics of human cortical neural progenitors and neurons [18–20]. FOXP1 mRNA and protein were absent from the KO-1 organoids, whereas the FOXP1 protein was gone but *FOXP1* mRNA persisted in KO-2 organoids as expected (**Figs 1D, 1E, and S2A–S2C and S1 Data**).

FOXP1 is enriched in cortical progenitors early in the second trimester [13]. To capture FOXP1 expression reflective of the early second trimester fetal brain, we performed immunostaining on the organoids using RGC and EN markers at day (D) 25, 40, 60, and 100 after inducing organoid formation (**S1A Fig**). We found that at D25, the organoids primarily contained SOX2+ progenitors with a thin layer of TUJ1+ neurons, a pattern analogous to a pre-

plate stage (**S1A Fig**). By D40, TUJ1+, CTIP2+, and CUX1+ staining indicated the presence of deep-layer neurons situated adjacent to the progenitors in an organization reminiscent of a cortical plate (CP) (**S1B–S1E Fig**). At D60, the CP was more developed, and FOXP1 expression was greater in the CP cells than at D40. By D100 (month 3), FOXP1 expression is primarily in CP cells, overlapping with the TUJ1+ ENs, with little expression within the SOX2+ RGCs (**S1A and S1B Fig**). This pattern of expression is similar to FOXP1 expression in the late second trimester fetal brain [13]. Among the time points we examined, D40 appeared to be most analogous to early second trimester fetal brain development. At D40, the cortical-like structures in the brain organoids expressed proteins abundant in the dorsal telencephalic lineage (DTL) cells in a pattern specific to a developing mammalian neocortex, such as FOXG1 in the progenitors and ENs, TBR2 in the intermediate progenitor cells (IPCs) lining the VZ, and CTIP2, and CUX1 in the ENs adjacent to the VZ (**S1C–S1E Fig**). To capture when FOXP1 expression is abundant in the basal cortical progenitors that can give rise to upper layer neurons where FOXP1 is also ultimately expressed [13], we performed the majority of our experiments at D40 ± 1 to 2 days [2,21]. Other work has reported outer radial glial cells in human brain organoids at around this same time [22]. An advantage of using human brain organoids to examine FOXP1 function is the potential presence of bRGCs, which are not as abundant in rodents and do not express FOXP1 [13].

## Identification of bRGC subtypes using single-cell transcriptomics

To study the cell type–specific functions of FOXP1 during cortical development, we performed snRNA-seq on wild-type (WT) and FOXP1 KO organoids (**Fig 2A**). After ensuring quality control (**see Methods and S2D and S2E Fig**), we obtained 151,336 nuclei across 9 samples, with an average of 30,310 reads per nucleus (**S1 Table**); this was sufficient to resolve the major cell types in the human brain organoids at this stage. *FOXP1* mRNA was absent from the KO-1 organoids, whereas *FOXP1* mRNA persisted in KO-2 organoids as expected (**S2A–S2C Fig**). We observed FOXP1 expression in approximately 75% of the bRGCs in WT organoids (**S2C Fig**), which is relatively similar to what we observed based on ICC (**S1F Fig and S1 Data**). Based on our snRNA-seq clustering and annotation [23,24], the brain organoids exhibited cell types typically represented in the DTL (**S3A–S3C Fig**), which is consistent with other brain organoid studies [2,21,25]. We identified 2 bRGC clusters (clusters 17 and 25) (**Fig 2A and 2B**) based on the annotation (**S3C Fig**). We further confirmed the cell type identity of these 2 clusters by performing a Fisher's exact test against a list of bRGC-enriched genes derived from a previous study [26] (**Fig 2C**). These results are in line with recent single-cell studies showing multiple subtypes of bRGCs in the developing human cortex [27].

To focus on cortical cell types that express FOXP1, we removed nondorsal telencephalic EN clusters from our analysis (**see Methods**). Additionally, as our KO-1 and KO-2 models showed similar transcriptional profiles, we merged them into 1 new dataset (KO) for the rest of our analyses (**S4A and S4B Fig**).

## FOXP1 deletion leads to changes in the developmental trajectory of bRGCs

Next, we sought to examine the developmental trajectories of the major cortical cell types reflected in the gene expression sequences and how the progression would be different between the WT and FOXP1 KO organoids. We performed pseudotemporal ordering of the cells using Monocle 3 [28]. We designated the root cells—assigned "0" in both the pseudotime uniform manifold approximation and projection (UMAP) (**Fig 2D**) and the density plot (**Fig 2E**)—as those triple-positive for *SOX2*, *PAX6*, and *HES5* transcripts. The apical radial glial cells (aRGCs), which gives rise to all progenitors and neurons in the cortex, are known to

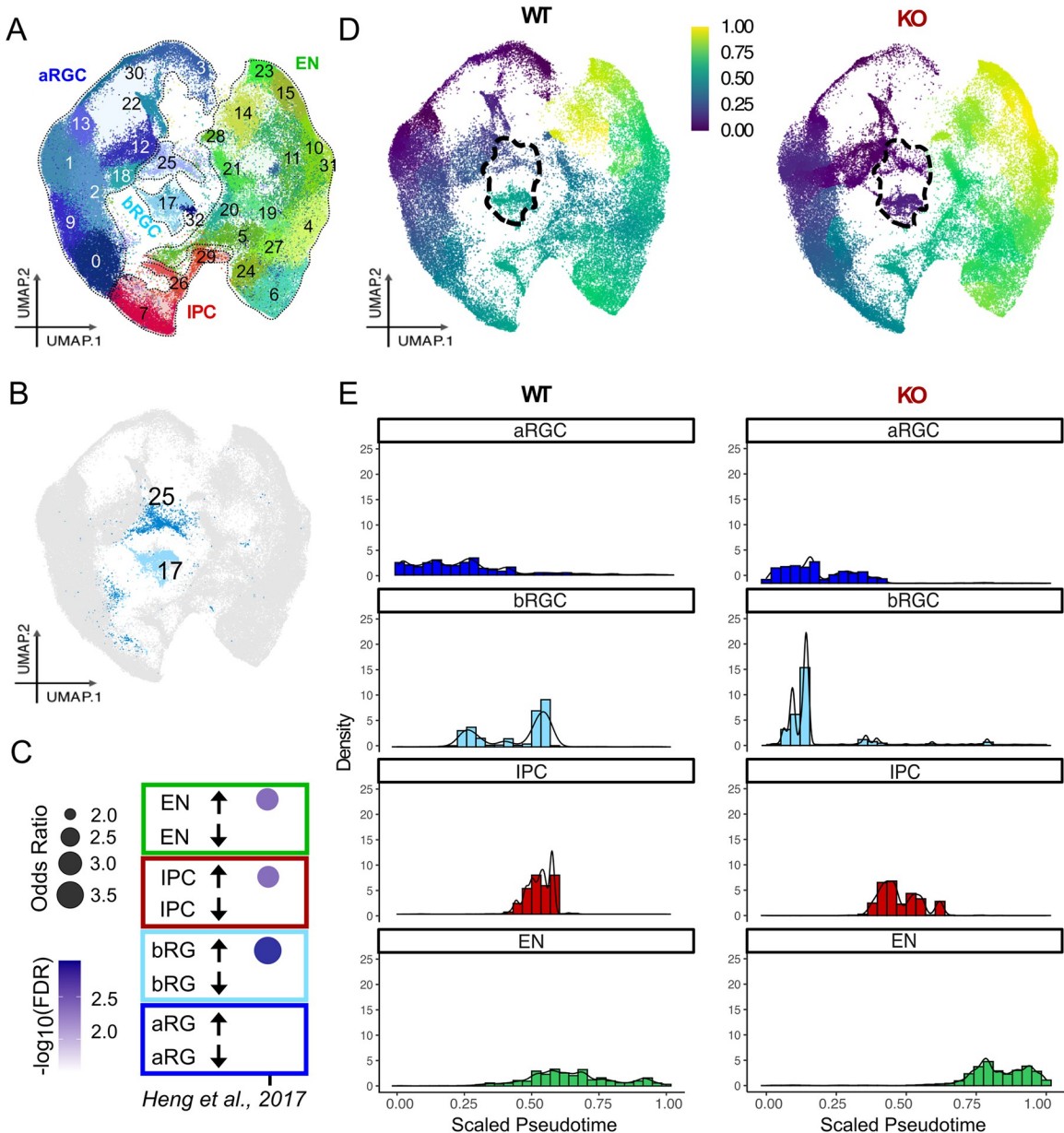

**Fig 2. snRNA-seq and pseudotime analysis of WT and FOXP1 KO organoids.** (**A**) Representation of diverse cell types in UMAP space. WT, KO-1, and KO-2 datasets are integrated and shown in the same UMAP. (**B**) A UMAP highlighting bRGC clusters 17 and 25. (**C**) Enrichment analysis of the bRGC genes [26] based on Fisher's exact test. (**D**) UMAPs with a scaled color scheme as a function of gene expression changes from the root cells ($SOX2^+PAX6^+HES5^+$). Changes in the basal progenitors are noted in a dotted line. WT (left) and the combined FOXP1 KO-1 and KO-2 datasets (right) are shown. (**E**) Density bar plots showing the number of pseudobulked cells (aRGC, bRGC, IPC, and EN) in the relative pseudotime scale. The root ("0") is same as in the above UMAP. aRGC, apical radial glial cell; bRGC, basal radial glial cell; EN, excitatory neuron; FOXP1, Forkhead Box P1; IPC, intermediate progenitor cell; KO, knockout; snRNA-seq, single-nuclei RNA-sequencing; UMAP, uniform manifold approximation and projection; WT, wild type.

express these TFs in the earliest stage of cortical development [29]. The end point, designated as "1," is the point at which no SOX2+PAX+HES5+ expression occurs and the greatest amount of transcriptional changes from the root cells is present. In the pseudotime UMAP, the relative changes in gene expression from the root cells are reflected in the color scale from one cluster to another (**Fig 2E**). bRGC clusters 17 and 25 showed the most striking color changes in the

KO compared to the WT (**Fig 2D, in dotted line**). Furthermore, when we plotted these changes using density plots, which show the number of cells in pseudotemporal order in a bar graph format, the bRGCs showed reduced gene expression changes from the root cells with the loss of FOXP1. We also observed a more skewed distribution of ENs in the KO (**Fig 2E**), which represents abundant dysregulation of gene expression programs and impaired neuronal differentiation. Additionally, we observed significantly decreased levels of some of the bRGC-enriched genes (i.e., *PTPRZ1*, *TNC*, *HOPX1*, *LIFR*, and *FAM107A*) [5,26] in the KOs (**S3D Fig**). These genes are known to be associated with proliferation and maturation of bRGCs [5].

Examination of bRGCs in KO1 and KO2 separately showed similar pattern of gene expression changes in the density plot (**S4B Fig**). We also saw similar patterns of gene expression changes when we examined bRGC subclusters #17 and #25 using both relative cell density and absolute cell numbers (**S4C Fig**). Other cell types, such as aRGCs and IPCs, showed subtle changes in the FOXP1 KO compared to WT (**Fig 2D and 2E**). Together, these results suggest either impaired differentiation or promotion of a more stem-like phenotype of bRGCs with loss of FOXP1. The bRGCs in the organoids with loss of FOXP1 may precociously differentiate into IPCs and contribute to the increased number of IPCs (**S6C Fig** and **S1 Data**) and to depletion of early progenitors. This, may in turn, lead to decreased numbers of ENs later on in development as reflected in our D100 organoids (**S7 Fig** and **S1 Data**).

## FOXP1 deletion leads to decreased proliferation and differentiation of bRGCs

We next wanted to assess how deletion of FOXP1 would affect the proliferation and differentiation of bRGCs at the protein level. First, we examined the number of bRGCs in both WT and FOXP1 KO. bRGCs are morphologically characterized by the loss of apical contacts and low expression of TBR2, an IPC marker [5,14,30–33]. We sparsely transduced organoids with an adenovirus-expressing GFP (AD-EGFP) to identify basally located cells without apical contacts (**Fig 3A**). We then costained the transduced organoids with an antibody to TBR2, an IPC marker, to more confidently label the bRGCs, since bRGCs have a relatively low expression of TBR2 compared to IPCs [5,14,32] (**Fig 3B**). While our criteria for selecting bRGCs may not capture all different types of bRGCs reported in the literature [34], we had at least observed bRGCs with 2 distinct gene expression profiles based on snRNA-seq (**Fig 2**). Using this costaining approach, we found fewer basally located EGFP+TBR2− cells without apical contacts in the KO (**Fig 3C** and **S1 Data**). To determine whether the decreased number of bRGCs was due to the reduced proliferative capacity of bRGCs, we also performed 5-bromo-2-deoxyuridine (BrdU) assays by pulsing the organoids at D40 with 100 μM BrdU for 2 hours and harvesting the samples 24 hours later (**Fig 3D**). We examined the organoids by immunostaining with a bRGC-enriched marker HOPX and the relative position of HOPX+ nuclei compared to adjacent cells (**Fig 3E–3G**). We observed fewer number of proliferating HOPX+BrdU + bRGCs among the total population of HOPX+ cells in the KO (**Fig 3H** and **S1 Data**). We supplemented this analysis by using another bRGC-enriched marker, PTPRZ, and found fewer PTPRZ+TBR2− bRGCs in the KO (**S5A and S5B Fig** and **S1 Data**). In addition to cellular proliferation, we also wanted to examine differentiation of bRGCs. Therefore, we counted TBR2+ cells located asymmetrically to HOPX+ cells, which represented neurogenic IPCs that we presume underwent indirect neurogenesis based on similar approaches in the field [35], and we observed fewer of these cells in the KO (**Fig 3I** and **S1 Data**). When we examined HOPX+ bRGCs located asymmetrically to CTIP2+Ens, which represent direct neurogenesis from bRGCs to Ens, we did not find significant differences between the genotypes (**S5C–S5E Fig** and **S1 Data**). Together, our data show that loss of FOXP1 may reduce the proliferative

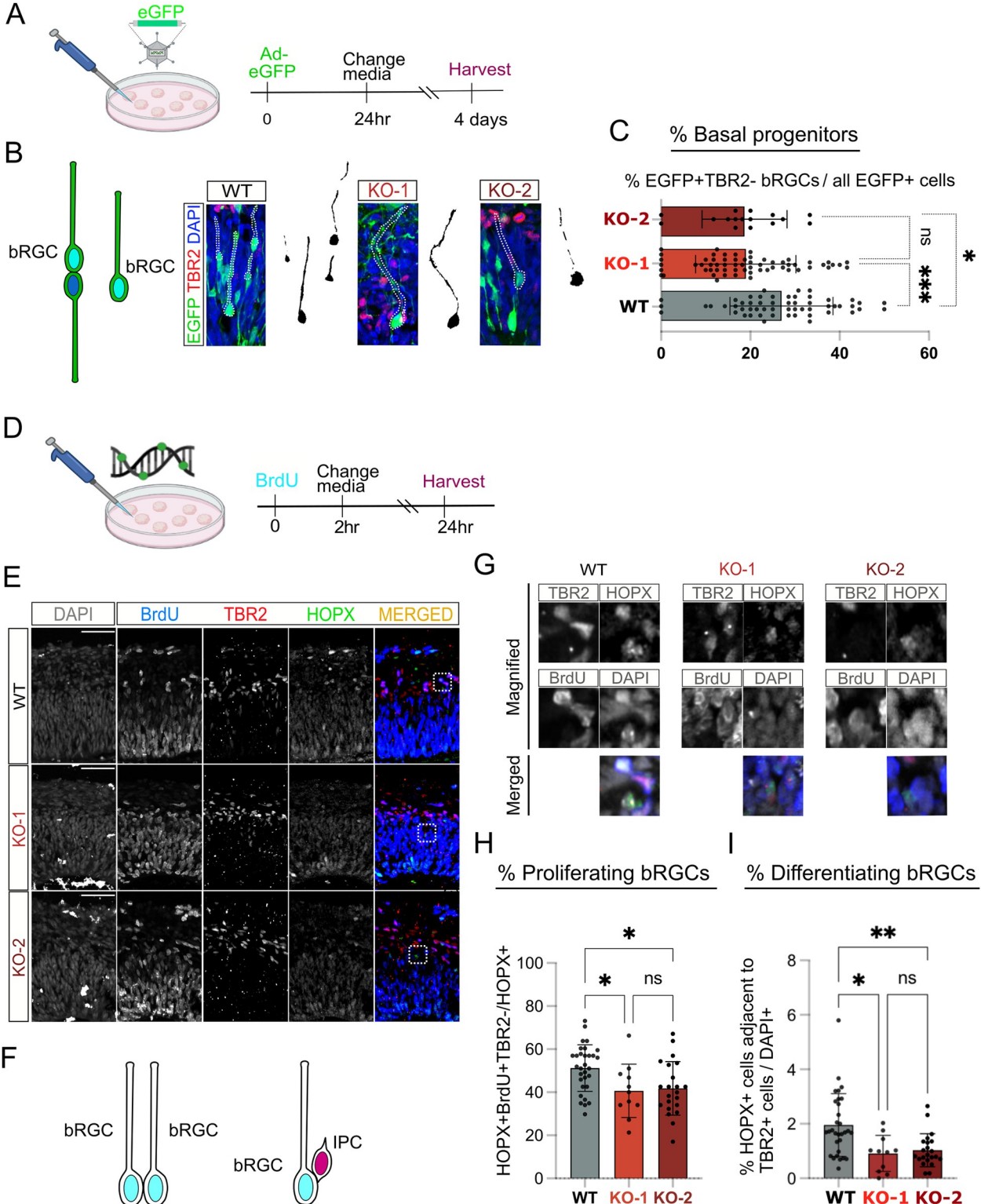

**Fig 3. Loss of FOXP1 results in decreased number of bRGCs.** (**A**) Schematic showing the design of the "Ad-EGFP" virus infection experiment. (**B**) Representative images of immunostaining performed on the "Ad-EGFP"-infected organoids at week 6 showing detection of bRGCs by morphology including loss of ventricular contact. (**C**) Quantification of progenitors that are without apical attachment and are TBR2−. (**D**) Schematic showing the design of the BrdU assays. Incubation with 100 μM BrdU for 2 hours was followed by media change, 24-hour incubation, and harvesting the organoids for fixation. (**E**) Representative images of immunostaining performed on BrdU-treated organoids at week 6. (**F**)

Schematic diagrams showing bRGC divisions. (**G**) Magnified images showing overlap between different markers. (**H**) Quantification of HOPX + dividing bRGCs. (**I**) Quantification of HOPX+ basal progenitors going through neurogenesis via IPCs. For all quantifications, *n* = 3–8 organoids per sample per genotype were used. In each organoid, 3–8 cortical structures with clear lamination patterns were examined. Data are represented in bar graphs as mean ± STD with individual data as dots; n.s. means $p > 0.05$, \**p* < 0.05, $p^{**} < 0.01$, and \*\*\**p* <0.001, Kruskal–Wallis ANOVA test with Dunn's multiple comparisons test as a post hoc was used. Scale bar = 100 μM. The numerical values that were used to generate the graphs can be found in S1 Data. Cartoon images were created with BioRender.com. Ad-EGFP, adenovirus-expressing GFP; BrdU, 5-bromo-2-deoxyuridine; bRGC, basal radial glial cell; FOXP1, Forkhead Box P1; IPC, intermediate progenitor cell; KO, knockout; WT, wild type.

capacity of bRGCs and also negatively affect indirect neurogenesis from bRGCs to IPCs, but not direct neurogenesis from bRGC to ENs. We did not observe changes in the number of neurons or the number of EN cells that exited the cell cycle to become neurons at this stage (**S6C, S6F and S6G Fig** and **S1 Data**).

We then assessed how a reduction in bRGCs during the early phase of neurogenesis might affect later neurogenesis. We cultured the organoids for approximately 3 months (100 days) and examined the number of neurons produced. In month 3, the progenitor pool becomes much smaller, as the majority of progenitors have differentiated into neurons (**S1A and S2B Fig**). In later cortical development stages, from mid-corticogenesis to early postnatal stages, FOXP1 is primarily expressed in postmitotic ENs in the CP and overlaps significantly with expression of SATB2, which is an abundant marker of ENs in cortical layers 2 to 5 [12,36]. We found that FOXP1 expression in our WT organoids at D100 also strongly overlaps with SATB2+ ENs (**S7A Fig**). When we examined whether loss of FOXP1 would impact these SATB2+ ENs (**S7B Fig**), we observed that there were significantly fewer SATB2+ ENs with the loss of FOXP1 (**S7C Fig** and **S1 Data**). The proportion of IPCs compared to the number of ENs was not statistically significantly different in this later stage of brain organoid development (**S7F Fig** and **S1 Data**). These results indicate that impaired neurogenesis in early cortical development with the loss of FOXP1 ultimately results in fewer neurons overall in later cortical development in this human model system.

## FOXP1 regulates NDD-associated genes

We wanted to determine which gene expression changes were potentially responsible for the shift in the developmental trajectory of bRGCs that we observed in the pseudotime analysis. Loss of FOXP1 led to a substantial number of differentially expressed genes (DEGs) in each cell type with progenitors having the greatest number of DEGs among the 4 major cell types (**Fig 4A and S2 Table**). Among the DEGs, we were specifically interested in neurogenesis and NDD-relevant genes that are dysregulated with the loss of FOXP1 in bRGCs. Our previous study indicated that FOXP1 regulates genes that are associated with different types of NDDs such as ASD, ID, and Fragile X Mental Retardation Protein (FMRP) target genes [37]. Therefore, we carried out a disease gene enrichment analysis that examined the overlap between our DEGs and ASD-, ID-, and FMRP-associated genes (**see Methods**) in a cell type–specific manner. The FOXP1 DEGs showed the greatest overlap with ASD genes in the bRGCs compared to the other cell types (**Fig 4B and S3 Table**). This indicates that with the loss of FOXP1, genetic vulnerability to ASD is most prevalent in the cell type that is linked to human cortical expansion in the early stage of corticogenesis.

To examine how FOXP1 regulates gene expression in different subtypes of bRGCs, we examined DEGs that are up- or down-regulated by FOXP1 in either the bRGC cluster 17 or 25 (**Fig 4C and S4 Table**). The gene ontology analysis for each subcluster of bRGCs showed that neurogenesis and projection neuron formation–related terms are commonly dysregulated in both clusters with loss of FOXP1 (**Fig 4D**), which indicates that FOXP1 regulates projection neuronal differentiation in bRGCs. Cluster 25, in particular, showed down-regulation of genes

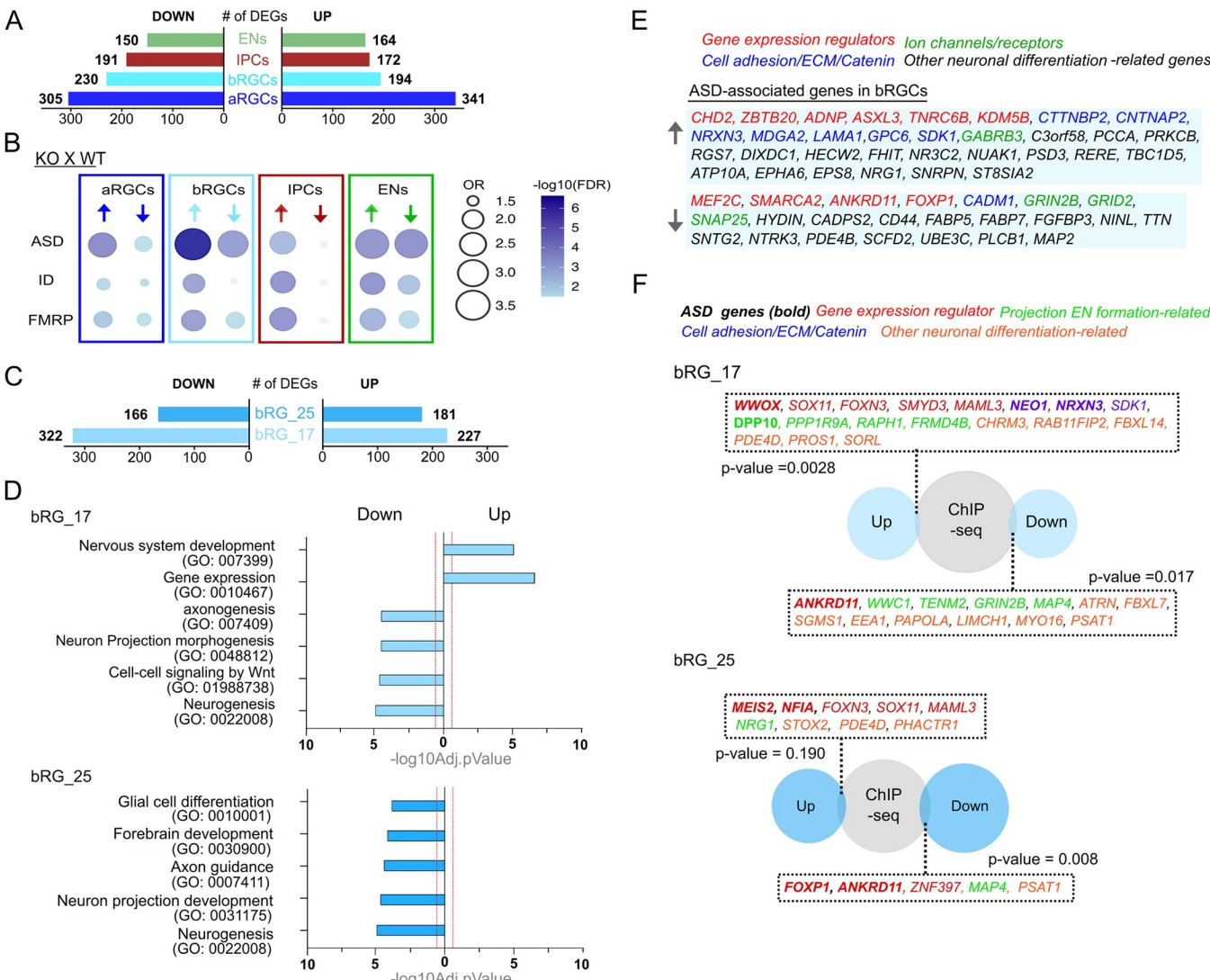

**Fig 4. Differential expression analysis of bRGC subtypes. (A)** Number of DEGs in each cell type. Significant DEGs are defined as log2(FC) ≥ 0.25, FDR ≤ 0.05. **(B)** Dot plots showing the enrichment of NDD-relevant DEGs in aRGC, bRGC, IPC, and EN cell types. **(C)** Number of DEGs identified in bRGC clusters. (log2(FC) ≥ 0.25, FDR ≤ 0.05) **(D)** Summarized biological process GO terms for up-regulated and down-regulated DEGs for each bRGC cluster. Significantly changed gene categories are defined as Benjamini–Hochberg FDR ≤ 0.05 (dashed line in red). **(E)** Lists of ASD-associated genes enriched in bRGCs. **(F)** Lists of bRGC enriched genes in both clusters that are differentially regulated by FOXP1. Scaled Venn diagrams for each cluster shows the overlap between the DEGs in bRGCs and a previously published chromatin immunoprecipitation assay with sequencing (ChIP-seq) dataset from our laboratory [37]. Fisher's exact test was performed to calculate statistical significance. ASD = high confidence ASD genes with score 1–3 from SFARI, ID = intellectual disability relevant genes, and FMRP = Fragile X relevant genes. aRGC, apical radial glial cell; ASD, autism spectrum disorder; bRGC, basal radial glial cell; ChIP-seq, chromatin immunoprecipitation sequencing; DEG, differentially expressed gene; FDR, false discovery rate; FMRP, Fragile X Mental Retardation Protein; FOXP1, Forkhead Box P1; EN, excitatory neuron; GO, gene ontology; ID, intellectual disability; IPC, intermediate progenitor cell; KO, knockout; NDD, neurodevelopmental disorder; OR, odds ratio.

associated with glial cell differentiation. When we examined individual genes, we found well-known ASD-relevant gene expression regulators, such as TFs (e.g., *MEF2C*, *ASXL3*, and *ZBTB20*), chromatin remodelers (e.g., *CHD2*, *ANDP*, *SMARCA2*, and *ANKRD11*), and an RNA-binding protein (e.g., *TNRC6B*) among the genes altered in the DEGs **(Figs 4E and S5)**. For example, the gene encoding, MEF2C, a TF involved in promoting neuronal differentiation and function [38,39], is down-regulated in bRGCs. Another gene, *ANKRD11*, which encodes a

chromatin regulator that controls transcription through histone acetylation and is implicated in cortical development, is down-regulated in bRGCs. In a previous study, KD of *ANKRD11* in both developing mouse cortical precursors at E12.5 and human embryonic stem cell (ESC)-derived cortical neural precursors led to decreased proliferation and differentiation [40].

Catenins (i.e., *CTTNBP2*), extracellular matrix (i.e., *LAMA1*), and cell adhesion–associated genes (i.e., *NRXN3; SDK1; CNTNAP2*) were other notable gene categories regulated by FOXP1 in the bRGCs (**Fig 4E** and **S4 and S5 Tables**). Catenins, extracellular matrix, and cell adhesion molecules interact with each other in the modulation of synapse structure formation, acquisition of growth factors, functioning of cell-signaling pathways, or synaptic communication with other cells [41,42]. In particular, *CNTNAP2* is a candidate gene for ASD and has been associated with human language development and cognition [9]. Previous studies have demonstrated that *CNTNAP2* is regulated by the FOXP family of TFs in multiple systems including cell lines and primary neurons [43,44]. Whole genome sequencing studies and gene manipulation studies have provided evidence supporting a regulatory role of FOXP1 on *CNTNAP2* [43,45]. However, to date, the regulation of *CNTNAP2* by FOXP1 in human bRGCs has not yet been examined. Our study provides the first evidence of the specific role of FOXP1 in regulating CNTNAP2 expression in human bRGCs, shedding new light on the cell type–specific regulatory mechanisms involved in cortical development and their potential implications in ASD.

To provide insights into the direct targets of FOXP1 in bRGCs, we overlapped the DEGs from each bRGC cluster and the DEGs from a previously published chromatin immunoprecipitation sequencing (ChIP-seq) dataset [37] from our lab (**Fig 4F**). Doing so yielded genes associated with diverse neurogenesis-related functions such as gene expression regulators, catenin/extracellular matrix/cell adhesion molecules, and projection neuronal formation in both clusters. The majority of the genes pertaining to those same broadly categorized functions were different between the 2 subclusters of bRGCs, which may indicate that FOXP1 regulates neurogenesis in the bRGCs through different mechanisms in each bRGC subtype. Additionally, our analysis showed that FOXP1 may directly regulate high confidence ASD genes (i.e., *NRXN3*, *NRG1*, and *SDK1* for up-regulation; *ANKRD11* and *GRIN2B* for down-regulation), early in the differentiation of bRGCs. These genes are important in a diverse range of projection neuronal formation such as neuronal proliferation, differentiation, migration, and function [46–50].

## Discussion

The developing human neocortex harbors highly heterogeneous cell types that are generated from seemingly homogenous progenitors under strict spatiotemporal control. bRGCs specifically play a vital role in the development of the human cortex. The dynamic changes that occur in bRGCs are initiated by regulatory elements, such as TFs. We used human cerebral organoids to model the production of bRGCs in a developing human cortical model. Because bRGCs are less common in rodent brain and do not express FOXP1 [13], we focused on bRGCs with respect to FOXP1 manipulation taking advantage of the relatively high percentage of bRGCs that express FOXP1 in the human brain organoids (>75%). Although bRGCs are a relatively rare cell type within the organoid model at D40 (approximately 4%), use of the D40 time point allows us to capture the earliest role for FOXP1 in directing bRGC function when cortical lamination is still very clearly articulated. Thus, we examined the dynamic gene orchestration governed by FOXP1 in bRGCs. We used human cerebral organoids to model the production of bRGCs in a developing human cortical model. Using snRNA-seq, we distinguished the subtypes of bRGCs based on their gene expression signatures. We highlighted the

altered developmental trajectory of bRGCs with the loss of FOXP1, as well as several neurogenesis- and NDD-related genes and signaling pathways. By immunostaining using cell type–specific markers, EGFP labeling, and BrdU assays, we were able to observe bRGC production, the reduction of bRGC to IPC and EN differentiation, and the reduced production of ENs in FOXP1 KOs. In addition to the regulation of neurogenesis by FOXP1 in bRGCs, another potential mechanism contributing to the reduction in neuronal numbers could be through indirect neurogenesis mediated via aRGCs and IPCs. Based on immunostaining for SOX2 +RGCs, TBR2+IPCs, and CTIP2+ENs, we observed premature transition from IPCs to ENs with loss of FOXP1 (**S6B–S6E Fig** and **S1** Data). This early depletion of progenitors may have led to the observed reduction in neurons. Additionally, we observed a trend in the number of SOX2+ RGCs being lower in the KOs compared to WTs, which denote the majority of aRGCs (**S6C Fig** and **S1** Data), and we saw changes in the division angle at the basal side where aRGCs divide (**S6A Fig** and **S1** Data). We observed that aRGCs at the basal side tend to divide more asymmetrically (oblique) with loss of FOXP1 compared to the WT, which is consistent with what had been shown in E13.5 mouse embryonic cortex [13]. Our interpretation of this result is that this increased asymmetric, oblique division may indicate division to generate 2 daughter cells of different cell fate such as IPCs or neurons [51,52]. While this shift in the developmental trajectory of IPCs was not reflected in the snRNA-seq pseudotime analysis, it would be important to know how FOXP1 regulates other basal progenitor cell types with human-specific modifications. Importantly, our experiments cannot fully distinguish whether the observed alterations in bRGCs are due to effects solely in bRGCs or from loss of *FOXP1* in aRGCs that give rise to bRGCs.

It has been suggested that one mechanism by which FOXP1 regulates cortical projection neurogenesis is through regulation of adhesion molecules [53]. Catenins often interact with the extracellular matrix or cell adhesion molecules in the modulation of synapse structure formation, acquisition of growth factors, functioning of cell-signaling pathways, or synaptic communication with other cells [41,42]. *CTNND2*, a gene encoding for δ-Catenin that is involved in WNT signaling through its interaction with N-cadherin as well as neurite outgrowth and function [54–57], is up-regulated in EN clusters. Since the loss of *Foxp1* leads to impaired dendritic and axonal formation [11,12], we asked whether elevated levels of CTNND2 would result in a similar phenotype as the loss of FOXP1 in the cortical neurons. Therefore, we performed a rescue experiment on the neuronal morphogenesis–related phenotypes by applying a CRISPR inhibition (CRISPRi) approach to *CTNND2* (**S8A and S8B Fig** and **S1 Data**). Using a published protocol [58], we generated 2D cortical neurons that express TUJ1, MAP2, and CUX1 (**S8C and S8D Fig**), which are the markers that are expressed in FOXP1+ cortical neurons during the mid-late neurogenesis as well as postmitotically. FOXP1 KO cells exhibited significantly fewer developed dendritic branches compared to WT. Inhibiting CTNND2 using CRISPRi rescued this phenotype to a significant level (**S8E and S8F Fig** and **S1 Data**). We believe this result could provide us with a better understanding of how FOXP1 orchestrates neurogenesis by regulating WNT signaling in the developing brain.

FOXP1 is a high-confidence ASD gene *[59]*. We found that FOXP1-regulated DEGs have a stronger association with ASD genes in bRGCs than other cell types. This indicates that, with a loss of FOXP1, genetic vulnerability to ASD is prevalent in a cell type linked to human cortical expansion in early corticogenesis. In addition to genes involved in proliferation and differentiation mentioned above, we also discovered DEGs related to neuronal functions, such as synaptic transmission and intrinsic excitability, in the bRGCs (**Fig 4D–4F**). These genes encode for glutamate receptors (e.g., *GRIND2B* and *GRID2*) and potassium (e.g., *KCNB1*) and chloride (e.g., *GABRB3*) ion channels. Previous studies have reported regulation of neuronal function in adult mouse hippocampal CA1 neurons and striatal spiny projection neurons by FOXP1

[37,60]. Besides neuronal firing, neuronal function–related genes such as neurotransmitter receptors and ion channels are known to play important roles for proliferation and differentiation of progenitors [61–64]. Our results suggest that FOXP1 primes neurogenic progenitor cells to proliferate as well as to differentiate into functional neurons.

Using currently available 3D brain organoid protocols, we successfully recapitulated many important characteristics of the developing cortex, such as the initial formation of cortical basal progenitors, followed by the differentiation of ENs into a layered fashion reminiscent of an early CP that is comprised of deep layer neurons. Mouse models with complete Foxp1 KO are embryonic lethal due to a heart defect [65]. However, FOXP1 KO brain organoids are free of those organismal complications and can be used to elucidate the essential contributions of FOXP1 to brain development effectively. However, while 3D brain organoids generate distinct ventricular-like zones and a primordial CP, they do not develop an elaborate 6-layered CP that contains mature ENs, in part due to the absence of inputs from other parts of the organoid. Therefore, it is challenging to study the alteration of the organization of ENs residing in the CP. Moreover, due to a lack of fully mature organized lamination and circuitry, studying how human FOXP1 affects radial migration as the CP expands remains to be elucidated.

In summary, our study has successfully shown that FOXP1 contributes to human-specific elements of the developing cortex such as bRGCs and governs a multitude of NDD-relevant neurogenesis pathways. These data provide opportunities for further exploration of the human-relevant gene expression driven by FOXP1 during brain development. In addition, validation of the key downstream target genes we identified can facilitate the development of therapeutics to treat FOXP1 syndrome and other forms of NDD.

## Methods

### Ethics statement

Stem cell work described in this manuscript has been conducted under the oversight and approval of the UT Southwestern Stem Cell Research Oversight (SCRO) Committee (Registration #8). UT Southwestern uses the International Society for Stem Cell Research (ISSCR) "Guidelines for Stem Cell Research and Clinical Translation" and NIH "National Institutes of Health Guidelines for Human Stem Cell Research" when establishing stem cell research standards. The male human iPSC line (WTC-11, cat #GM25256) was provided by Bruce R. Conklin (The Gladstone Institutes and UCSF). The cell line was originally obtained under informed consent and was provided in deidentified form to the authors of this study and therefore not considered human participants research for the purpose of this study.

### Stem cell culture

A male human iPSC line (WTC-11, cat #GM25256) was provided by Bruce R. Conklin (The Gladstone Institutes and UCSF). The cell lines were verified to have normal karyotype based on G-banding technique, free of mycoplasma contamination, and were kept under passage number 10 for differentiation. Proliferation markers (OCT4, SOX2, KLF4, and NANOG) were checked by qPCR or immunostaining. The cell lines were cultivated in mTeSRTM1 medium (cat 85870, STEMCELL Technologies) using feeder-free culture protocols in 6-well plates (cat #4936, Corning) that were coated with growth factor–reduced Matrigel matrix hESC-qualified (cat #354277, BD Biosciences). Cells were passaged 1:4 every 3 to 4 days, using Gentle Cell Dissociation Medium (cat #07174, STEMCELL Technologies), and ROCK inhibitor was added at a final concentration of 10 μM (cat #50-863-6, Fisher Scientific) for the first 16 to 20 hours of passage. The cells were maintained with daily medium change without ROCK inhibitor.

## CRISPR-Cas9–edited iPSC line generation

The FOXP1-KO cell lines using strategy 1 (KO-1) were generated by targeting the 5′ and 3′ ends of *FOXP1* with a pair of sgRNAs. For 5′ prime UTR targeting, the following set of sgRNAs were used: 5′-CACCGACAAACTTTCGGGGTTCCCGC-3′ and 5′- AAACGCGGGAACCCG AAAGTTTGTC-3′. For 3′ prime UTR targeting: 5′-CACCGCATCTTACAAGACGGACTC T-3′ and 5′- AAACAGAGTCCGTCTTGTAAGATGC-3′.

We identified putative 5′ and 3′ ends as ±1,000 bp of the first and last exons of *FOXP1* iso-form 1 (OMIM: 605515). To mitigate potential off-target effects of gene editing, sgRNA candidates were analyzed using the online CRISPR Design tool developed by the Zhang laboratory (http://crispr.mit.edu/), and the sgRNA sequences with fewest off-target sites in the human genome were selected for use. Each sgRNA was inserted into "pSpCas9(BB)-2A-GFP (PX458)," which was obtained from Addgene (Plasmid #48138). Cells were incubated with ROCK inhibitor for 1 to 3 hours and then electroporated with the pair of sgRNAs. About 48 to 72 hours post electroporation, cells were subjected to fluorescence-activated cell sorting (FACS) for GFP+ signals and replated as single clones for the subsequent 10 to 12 days. The second FOXP1 KO (KO-2) cell lines were generated using the CRISPR Paint strategy [66]. We used "pCRISPaint-TagGFP2-PuroR" plasmid (#80970) to insert "TagGFP2-2A-PuroR-PolyA" into the *FOXP1* locus. (Upon differentiation into organoids, the GFP signal is expressed at low levels in the KO-2 cells and requires the use of a TagGFP2-specific antibody.) Two sgRNAs were designed to target exon 11 or 14. For exon 11 targeting, the following set of sgRNAs are used: 5′ CACCGGTCCATTGGTAGAGGCATGT-3′ and 5′-AAACACATGCCTCTACCAAT GGACc-3′. For exon 14 targeting, the following set of gRNAs are used: 5′- CACCGGTAAGT ATTGATCCCCACCA -3′ and 5′- AAACTGGTGGGGGATCAATACTTACc -3′. sgRNA, selector, and GFP donor DNA were electroporated in a 1:1:3 molar ratio using a 4D Lonza electroporator (cat # V4XP-3024, Lonza).

Cells were subjected to puromycin selection (0.3 to 0.5 μg/ml) following the method from a published study [67]. To verify deletion of *FOXP1*, positive clones from both strategies were confirmed via Sanger sequencing, qPCR, immunostaining, and western blot.

## Brain organoid generation

Brain organoid generation was performed using a modification of a commercially available cerebral organoid kit (cat # 08571, Stem Cell Technologies) together with the method published from a published study [20]. After day 40, we used the maturation medium following Lancaster and colleagues [20]. Brain organoids were kept on an orbital shaker (cat # BT4500, Benchmark) starting on D11 at 79 rpm. Brain organoids were analyzed at 40 days in vitro (DIV) for proper cortical lamination using the following markers: SOX2, PAX6, and HOPX for NSC and RG, TBR2 for IPC, CTIP2 and TBR1 for pre-plate or deep layer neurons, TUJ1 for immature neurons, and FOXG1 for dorsal telencephalic neural cells.

## RT-qPCR

Organoid RNA was extracted following the protocol supplied with RNeasy Total RNA Kit (cat #533179, Qiagen) and TRIzol reagent (cat # 15596018, Thermo Fisher Scientific). The extracted RNA was reverse transcribed following the protocol supplied with SSIII First-Strand Super Mix for real-time PCR (RT-PCR) (cat # 18080–400, Invitrogen Life Technologies). Quantitative real-time PCR (qPCR) was carried out using the CFX384 Real-Time System (Bio-Rad). Reactions were run in triplicate or quadruplicates, and expression of each gene was normalized to the geometric mean of 18s and *β*-actin as housekeeping genes and WT values to generate ΔΔCT. The primer sequences of each gene can be provided upon request.

## Western blot

Western blotting was performed as previously described in a published study from our laboratory [36]. Individual organoids were lysed in radioimmunoprecipitation assay (RIPA) buffer containing protease inhibitors. Protein concentrations were determined through a Bradford assay (Bio-Rad Laboratories). Approximately 30 to 50 μg of proteins for each of the genotypes were run on an SDS-PAGE gel for 2 hours at room temperature and transferred to an immune-Blot PVDF Membrane (Bio-Rad Laboratories) for 16 hours at 4˚C. Blots were imaged using an Odyssey Infrared Imaging System (LI-COR Biosciences). GAPDH (cat #MAB374, Milipore, 1:1,000 dilution), FOXP1 (cat #2005S, Cell Signaling, 1:1,000 dilution), and FOXP1 [68] (1:1,000 dilution) antibodies were used.

## Immunocytochemistry (ICC) staining

The organoid samples were fixed with 4% paraformaldehyde at 4˚C overnight on a shaker. The next day, the organoids were washed 3 times for 5 minutes with PBS and transferred to 30% sucrose for another overnight incubation at 4˚C. Then, the samples were carefully embedded in Tissue-Tek CRYO-OCT Compound (cat #14-373-65, Thermo Fisher Scientific), which slowly solidified on dry ice. The frozen organoid samples were sectioned at 20 μM thickness with a cryostat and warmed to room temperature. Some antibodies were subjected to antigen retrieval using citrate buffer (10 mM tri-sodium citrate, 0.05% Tween-20 (pH 6)) for 10 minutes at 95˚C. Free aldehydes were quenched with 0.3 M glycine in TBS for 1 hour at room temperature. This was followed by blocking for 1 hour at room temperature in 1% (w/v) bovine serum albumin (BSA) and 5% (v/v) normal donkey serum (NDS) in TBST (150 mM NaCl, 10 mM Tris (pH 8), 0.05% Tween 20). The primary antibodies were diluted in the blocking buffer and incubated for 2 nights at 4˚C. Coverslips were then washed 5 times in TBS and incubated overnight at 4˚C with the relevant fluorescent-conjugated secondary antibody. Sections were washed 3 times, stained for DAPI for 5 minutes, and then washed twice more. Finally, coverslips were mounted using ProLong Diamond Antifade Mountant (cat #P35970, Thermo Fisher Scientific). Analysis of cell morphology and differentiation was performed across 3 separate batches of differentiation experiments, using at least 3 samples per experiment. Only sections from the middle of the organoids were used. WT and KO samples were sectioned and mounted onto the same slide so that the immunostaining condition were the same across different genotypes. Only DAPI[+] cells were analyzed. Antibodies used were as follows: FOXP1 (cat #2005S, Cell Signaling, 1:1,000 dilution), FOXP1 [68] (1:1,000 dilution), FOXG1 (cat #18259, Abcam, 1:500 dilution), SOX2 (ab3045S, Abcam, Cell Signaling, 1:500 dilution), SOX2 (cat #sc17320, Santa Cruz, 1:500 dilution), PAX6 (cat # HPA030775, Sigma Aldrich, 1:1,000 dilution), EOMES (cat # AF6166SP, Fisher Scientific, 1:500 dilution), EOMES (cat #ab23345, Abcam, 1:500 dilution), HOPX (cat #11419-1-AP, Proteintech, 1:100 dilution), TBR1 (cat #ab31940, Abcam, 1:100 dilution), CTIP2 (cat # ab18465, Abcam, 1:500 dilution), PTPRZ1 (cat #HPA015103, Sigma Aldrich, 1:500), BrdU (cat # MA5-11285, Thermo Fisher, 1:500 dilution), BrdU (cat #ab6326, Abcam, 1:1,000 dilution), GFP (cat #600-101-215, Rockland, 1:1,000 dilution), GFP (cat #GFP-1010, Aves, 1:1,000 dilution), pVimentin (cat # 50-459-01, MBL International, 1:1,000 dilution), and SATB2 (cat #51502, Abcam, 1:500 dilution).

## Microscope imaging and analysis

Images were generated using a Zeiss confocal laser scanning microscope (LSM880). Six z-stack images for samples with 16-μM thickness were collected using a 20× lens within a 1,024 × 1,024 pixel field of view across all images and averaged per section. All the results were quantified either manually using ImageJ (http://imagej.nih.gov/ij) or automatically using

CellProfiler (http://cellprofiler.org). Differences between genotypes or conditions were assessed using either the combination of *t* test and Kolmogorov–Smirnov test or a one-way ANOVA with multiple comparisons and Brown–Forsythe and Welch ANOVA tests. Graph-Pad Prism software was used to perform statistical tests and obtain *p*-values. Sample size for each experiment is indicated in the figure legends.

### Analyses of cortical neural cell distribution

Analyses of the number of different cell types—RGCs, IPCs, ENs, or any combination of their overlaps—were performed following previously described methods [20,69] with some modifications. The modifications are as follows: For staining, we used the middle 8 to 10 sections of each organoid at 20 μM thickness. Since the size and shape of the cortical structures varied within the same organoid as well as across different ones, we took images of all the cortical structures whose shape did not overlap or fuse with another cortical structure and were positive for dorsal telencephalic markers unbiasedly. In addition to the cell type–specific markers, well-differentiated organoids show individual cortical structures with a clear presence of a ventricular space, a VZ that is full of radially organized cells, and a dense CP that is separated from the VZ by a cell-sparse zone. We drew a rectangular ROI (477 × 238 micrometer) from the basal surface to the apical surface and quantified the number of cells either manually using FIJI or automatically using CellProfiler for each image. ROI drawing and quantifications were performed in a bias-free manner, and the same quantification method was used across the genotypes in every experiment throughout the data generation. For each experiment, we used *n* = 3–8 WT, KO-1, and KO-2 organoids that were generated from the isogenic iPSC lines.

### AD-eGFP infection and bRGC quantification

AD-eGFP (cat#106, vector lab) was added to media at 1:2,000 and incubated for 24 hours. The virus was added at or around D40 (±2 to 3 days) and incubated for 24 hours, and organoids were collected 72 hours later. The organoids were then stained with anti-GFP (cat #AB011, Evrogen, 1:1,000 dilution) for better visualization of GFP-infected cells. bRGCs were identified based on distinct morphology and the absence of the IPC marker TBR2 [35,70–72].

### BrdU labeling proliferation and differentiation assays

Cell proliferation and differentiation rates were determined by labeling with BrdU (cat # B5002, Sigma-Aldrich). The organoids were pulsed with a single dose of 100 μM BrdU for 2 hours, washed 3 times with PBS, and either harvested right away or chased in the organoid medium without BrdU for 24 hours. Anti-BrdU (cat# ab6326, Abcam or cat# MA5-11285, Thermo Fisher Scientific, 1:400 dilution) was used in conjunction with anti-EOMES (cat #50-4877-42 Thermo Fisher Scientific or cat# AF6166SP Thermo Fisher Scientific, 1:400 dilution) or anti-CTIP2 (cat #ab18465, Abcam, 1:400 dilution) to examine proliferating cells in active S-phase or differentiating cells that had just finished the last division to become neurons, respectively.

### Cell cycle exit analysis

In conjunction with BrdU, which marks cells going through S-phase, anti-Ki67 (cat #ab15580, Abcam) was also used to examine cells that are in all stages of the cell cycle (S, G1, M, and G2). The difference between $BrdU^+$ and $Ki67^+$ cells ($BrdU^+Ki67^-$) was used to examine proliferating or differentiating G-M cell population, which reflected the number of cells exiting the cell cycle.

## Neuronal morphology rescue experiment using CRISPRi strategy

WT and KO iPSC lines were differentiated into 2D induced neurons based on published protocols with some optimizations necessary for our own iPSC lines [58,73]. The modifications are as follows: We coated the plates with Poly-O-Laminin the night before and then coated the plates with Matrigel for 1 hour at 37˚C prior to plating the cells on glass coverslips. We fed the cells Laminin a concentration of 5 μg/ml every 2 to 3 days. The concentration of Puromycin varied depending on the stage of differentiation, and we typically used somewhere between .3 and 1 μg/ml. Reverse tetracycline-controlled transactivator (rtTA) (FUdeltaG@-rtTA; Addgene, #19780) and NGN (pTet-O-Ngn2-Puro; Addgene, #52047) were purchased from Addgene. Lentivirus-infected cells were selected for expression of the puromycin-resistant gene. Cells were plated, differentiated, and maintained on glass coverslips in 24-well plates for up to 28 days. Rescue experiments using CRISPRi, which was designed to inhibit the expression of CTNND2 using dCas9, began on day 6 when the cells were transitioning from progenitor medium to differentiation medium and were examined on day 28. Targeting cells with CRISPRi was done sparsely with a GFP reporter downstream of gRNA to examine the morphology of neurons thoroughly. sgRNA pairs were designed using CRISPick [74,75]. F: 5′-ACCGGTCCAGGGCGTGCGTTCCCA-3′ and R: 5′- AACTGGGAACGCACGCCCTGG ACC-3′ were designed to be 131 bp away from the known TSS of CTNND2. The sgRNA sequence with the top 4 fewest off-target sites in the human genome was selected for use. The pair of sgRNA was cloned into pAAV-U6-CMV-GFP (Addgene, #R0569). All the virus packaging work was performed by Neuroconnectivity Core at Baylor College of Medicine Intellectual and Developmental Disabilities Research Center. To ensure successful neuronal differentiation, we checked for PAX6, Tuj1, MAP2, SATB2, vGLUT2, and CUX1 expression on days 5, 14, and 28. All the WT and KO cells were grown under the same conditions, and dendritic complexity analysis was performed following a published method [76] across both WT and KO cells.

## Nuclei isolation and library generation

Nuclei isolation was performed following published methods [77,78]. snRNA-seq was performed using a 10x Genomics Chromium system following published methods [77,78]. Cortical regions of the organoids were microdissected using the following method [25]. The organoids used in these experiments were $n = 3$ for each genotype: iPSC-derived WT, FOXP1 KO-1, and FOXP1 KO-2. Approximately 10,000 nuclei per sample per genotype were targeted for the experiment. Droplet-based snRNA-seq libraries were prepared using the Chromium Single Cell 3′ v3 kit (cat #120237 10x Genomics) according to the manufacturer's protocol and were sequenced using an Illumina Nova-Seq 6000 and NEXT-seq 500 for a total of over 3.5 billion reads.

## Sequence alignment and counting

Raw sequencing data were acquired from the North Texas Genome Center at the University of Texas at Arlington and McDermott Sequencing Core at UT Southwestern in the form of binary base call (BCL) files. Raw BCL files were then demultiplexed with 10x Genomics i7 indices (used during library preparation) using Illumina's bcl2fastq v2.19.1 and *"cellranger mkfastq"* from 10x Genomics CellRanger v3.0.2 tools. Extracted paired-end reads (28 bp long R1–16 bp 10x cell barcode and 12 bp UMI sequence information, 124 bp long R2—transcript sequence information from cDNA fragment) were first checked for read quality using FASTQC v0.11.5 (FastQC, Babraham Bioinformatics, URL: https://www.bioinformatics.babraham.ac.uk/projects/fastqc). Extracted paired-end reads were then aligned to the

reference human genome (GRCh38.p12) from University of California Santa Cruz (UCSC) genome browser and reference human annotation (Gencode v28) and counted using *"cellranger count"* from 10x Genomics CellRanger v3.0.2 tools. Since the nuclear transcriptome contained unspliced transcripts, reads mapping to a pre-mRNA reference file were counted. The resulting raw UMI count matrix contains genes as rows and nuclei as columns and was further used for downstream analysis.

## Clustering and cell type annotation

Raw and combined UMI counts for a total of 151,336 nuclei (62,899 nuclei for WT, 58,466 nuclei for KO, and 29,971 nuclei for KO2) were used for clustering using the Seurat R analysis pipeline (from 10X Genomics CellRanger v3.0.1 tools). For each genotype, nuclei with more than 15,000 molecules (number of UMIs per nucleus) and nuclei with more than 10% mitochondrial content were filtered out to discard potential doublets and unhealthy cells. Also, genes with no expression in any nucleus and genes from chromosomes X, Y, and M were removed. Seurat objects with filtered datasets for each genotype (59,041 nuclei for WT, 53,230 nuclei for KO, and 28,438 nuclei for KO2) were then normalized using "sctransform" and scored for cell cycle genes following Seurat guidelines. Individual SCTransformed and cell cycle scored Seurat objects were then integrated using the reciprocal PCA (RPCA) approach and clustered using the original Louvain algorithm. Clusters were visualized with UMAP [79,80] in 2 dimensions. A resolution of 1.2 was selected based on clustering stability using "clustree" R package [81]. Clusters were further annotated into a broad category of cell types using the gene markers enriched (FDR $\leq$ 0.05 and log2 (fold change) $\geq$ 0.25) in every cluster identified using "FindAllMarkers" and performing Fisher exact based enrichment against cell classes [23,24]. Further, only nuclei identified as from the DTL were retained by subsetting the Seurat object. Out of 36, 13 clusters of nontelencephalic or unknown identity were removed. Raw counts for a total of 116,607 DTL nuclei (50,533 nuclei for WT, 40,028 nuclei for KO, and 26,046 nuclei for KO2) were then integrated using RPCA, followed by identifying clusters for DTL nuclei. Clustered nuclei were visualized using UMAP. A resolution of 1.6 was selected based on clustering stability using the "clustree" R package [81]. Clusters were further annotated into cell types using the gene markers enriched in every cluster identified using "FindAllMarkers" and performing Fisher exact based enrichment against cell classes [23,24]. The identification of bRGC clusters was determined by a Fisher exact based enrichment analysis of cluster of RG types with genes annotated as markers of bRGCs [26]. Clusters that were identified as nondorsal telencephalic cortical cell types and inhibitory interneurons based on our annotation method were removed from the analysis. We also further confirmed the removal of these cell types using the marker genes derived from a published study [23]. Out of 140,709 total number of nuclei, 24,102 nuclei and 7,191 nuclei that were annotated as nondorsal telencephalic cortical cell types and inhibitory neurons, respectively, were discarded from the final dataset that was used for downstream analysis.

## Pseudobulk differential expression analysis

Clusters were grouped into broad categories such as RG, IPC, and EN based on cell type markers and enrichment against the reference dataset using Fisher's exact test. For differential gene analysis, cells corresponding to WT or KO genotypes were grouped within each broad cellular category. Genes with altered expression in KO were then identified using a Wilcoxon test from Seurat v3 [82] FDR $\leq$ 0.05 and log2(fold change) $\geq$ 0.25. The functional annotation of DEGs was performed using the ToppGene Suite [83] with a background of 8,023 genes. The

background genes are genes that are expressed in either WT or KO across all cell types. Gene ontology categories with Benjamini–Hochberg FDR ≤ 0.05 were summarized using REVIGO [84].

### Pseudotime trajectory analysis

The filtered Seurat object for DTL nuclei without inhibitory neurons described above was first split into genotype-specific subsets for WT and KO and then converted into Monocle compatible objects using "*as.cell_data_set*" command. Genotype-specific subsets were then preprocessed (*cluster_cells*, *learn_graph*) using the standard Monocle pipeline. Nuclei with the greatest *SOX2*, *PAX6*, and *HES5* expression were then selected as a root population for performing pseudotime trajectory analysis (*order_cells*). UMAP plots colored by scaled pseudotime values were then generated accompanied by density plots and histograms corresponding to broad cell types (RG, IPC, and EN).

### ASD gene enrichment analysis

The enrichment of pseudobulk DEGs and ASD-relevant genes was performed using the Fisher exact test. Disease-relevant gene sets were used from a previous study [85]. Fisher exact tests were performed in R with the following parameters: alternative = "greater", conf.level = 0.85. Bubble dot plots were generated using odds ratio (OR) and Benjamini–Hochberg-adjusted *p*-values (FDR).

### Gene overlap analysis

Scaled Venn diagrams were made using https://www.biovenn.nl/. Gene overlap analysis was performed using Fisher exact test. DEGs from ChIP-seq experiments were obtained from a previous publication [37].

### Quantification and statistical analysis

For snRNA-seq transcriptomic data, nonparametric Wilcoxon rank-sum tests were used for differential gene analysis. The methods for differential gene expression using Seurat v3 are detailed in the "Pseudobulk differential expression analysis" section. The results of differential gene expression analyses are listed in S2, S3, and S4 Tables, and subsets of these comparisons are included in Figs 2 and 4.

For statistical analysis of IHC quantification, individual organoids were treated as biological replicates. Organoid samples were randomly taken from the culture for experiments and analyses. The sample sizes were designed to account for the variability between organoids within the same batch of differentiation and meet current standards in human brain organoid-related studies. Data analyses comparing WT and FOXP1 KO organoids were performed blindly or using an automated quantification method that was applied similarly across the different genotypes. Data are presented as mean ± SEM, or mean ± SD, unless otherwise indicated in the figure legends. Statistical analyses were performed using Prism software. The appropriate statistical tests for each experiment are stated in figure legends. Statistical significance was defined by *p*-value or adjusted *p*-values < 0.05.

To calculate enrichment of overlapping datasets, Fisher exact test was used. A Benjamini–Hochberg-adjusted *p*-value was applied as a multiple comparisons adjustment. The results of these tests are shown in Figs 2, 4 and S3.

For gene ontology enrichments, a one-sided hypergeometric test was used to test overrepresentation of functional categories. A Benjamini-Hochberg adjusted p-value was applied as a multiple comparisons adjustment, and the results are shown in Fig 4 and S5 Table.

## Supporting information

**S1 Fig. Brain organoid development over time. (A)** Organoid immunostaining at D25, D40, D60, and D100 showing expression of SOX2, TUJ1, and FOXP1. **(B)** ROI selected from panel (A) showing FOXP1 expression. **(C)** Overlap between cell type markers and FOXP1. FOXG1 marks DTL cells, TBR2 marks IPCs, and CTIP2 marks ENs. **(D)** Magnified images from panel (C) showing the overlap among cell type markers. **(E)** Detection of "ad-EGFP"-infected FOXP1+ bRGCs based on morphology, location, and absence of TBR2 expression. **(F)** Quantification of FOXP1+ aRGCs, bRGCs, IPCs, and ENs. Scale bar = 100 μM for panels (A) and (C), 50 μM for panel S1B and S1E Fig. *n* = 4 cortical structures used from WT organoids. The numerical values that were used to generate the graph can be found in S1 Data. Ad-EGFP, adenovirus-expressing GFP; aRGC, apical radial glial cell; bRGC, basal radial glial cell; DTL, dorsal telencephalic lineage; EN, excitatory neuron; FOXP1, Forkhead Box P1; IPC, intermediate progenitor cell; ROI, region of interest; WT, wild type.
(TIFF)

**S2 Fig. snRNA-seq quality control and FOXP1 expression in the CRISPR-edited organoids. (A)** FOXP1 expression level in each of the pseudobulked cell types, aRGCs, bRGCs, IPCs, and ENs per genotype. **(B)** Genome track files at the *FOXP1* gene showing the pileup of the sequencing reads in this region of the genome for each genotype. **(C)** Percentage of FOXP1$^+$ cells in individual clusters. **(D)** UMI count, number of detected genes, and percentage mitochondria gene in each cluster. **(E)** UMI count, number of detected genes, and percentage mitochondrial genes in each genotype. aRGC, apical radial glial cell; bRGC, basal radial glial cell; EN, excitatory neuron; FOXP1, Forkhead Box P1; IPC, intermediate progenitor cell; snRNA-seq, single-nuclei RNA-sequencing.
(PDF)

**S3 Fig. Representation of the diverse cell types profiled in the organoids. (A)** Feature plots showing distribution of cells across different genotypes. **(B)** Violin plots showing gene expression of markers representing the DTL versus other brain regions. **(C)** Dot plot showing gene expression correlation between our brain organoid dataset and a human fetal cortex scRNA-seq dataset from the second trimester of gestation [24]. Dotted lines in blue indicate bRGC cell clusters. **(D)** Violin plots showing the expression level of bRGC-marker genes [5] that are differentially regulated between WT and KO in each bRGC subcluster. Y-axis represents log10 (CPM) values. bRGC, basal radial glial cell; DTL, dorsal telencephalic lineage; KO, knockout; OR, odds ratio; scRNA-seq, single-cell RNA-sequencing; WT, wild type.
(PNG)

**S4 Fig. Similar gene expression pattern of KO-1 and KO-2. (A)** Correlation of DEGs between KO-1 and KO-2 organoids in all cells as well as each of the pseudobulked cell types, aRGC, bRGC, IPC, and ENs. **(B)** Density plots showing the number of pseudobulked cells (aRGC, bRGC, IPC, and EN) of KO-1 and KO-2 organoids separately. aRGC, apical radial glial cell; bRGC, basal radial glial cell; EN, excitatory neuron; DEG, differentially expressed gene; IPC, intermediate progenitor cell; KO, knockout.
(PNG)

**S5 Fig. bRGC-driven direct neurogenesis, quantification of PTPRZ+ bRGCS. (A)** Immunostaining showing PAX6, TBR2, and the bRGC marker PTPRZ in week 6 organoids. Scale bar = 50 μM **(B)** Quantification of bRGCs using PTPRZ+TBR2− bRGCs. **(C)** Schematic showing the experimental design of the BrdU treatment at week 6. Created with BioRender.com. **(D)** Immunostaining showing CTIP2, HOPX, BrdU, and DAPI in week 6 organoids. Scale bar = 50 μM **(E)** Quantification of neurogenic bRGCs, which are CTIP2+ cells located next to HOPX+ cells. These cells may represent neurons born directly from bRGC to EN. For all quantifications, $n$ = 3–8 organoids per sample per genotype was used. In each organoid, 3–9 cortical structures with clear lamination patterns were examined. Data are represented in bar graphs as mean ± STD with individual data as dots; n.s. = not statistically significant, *$p < 0.05$, Kruskal–Wallis ANOVA test with Dunn's multiple comparisons test as a post hoc. The numerical values that were used to generate the graphs can be found in S1 Data. BrdU, 5-bromo-2-deoxyuridine; bRGC, basal radial glial cell; EN, excitatory neuron. (PDF)

**S6 Fig. Deletion of FOXP1 results in increased numbers and protracted differentiation of IPCs. (A)** Representative images of symmetric and asymmetric division (left) and quantification of the divisions across WT, KO-1 and KO-2. **(B)** Representative images showing changes observed in layering with the loss of FOXP1 (left) and examples of ROIs selected from left panel (right). SOX2 expression marks RGCs, TBR2 marks IPCs and CTIP2 marks ENs. Scale bar = 100 μM for the left images and 50 μM for the ROI images on the right. **(C)** Quantification of RGCs, IPCs, and ENs. **(D)** Quantification of SOX2+ RGCs that express IPC marker TBR2. **(E)** Quantification of CTIP2+ ENs that express IPC marker TBR2. **(F)** Representative images of BrdU-treated brain organoids sectioned and stained for Ki67, BrdU, and CTIP2 expression. Scale bar = 50 μM **(G)** The difference between BrdU and Ki67 staining showing cell cycle exit rate in the CTIP2+ postmitotic neurons. For all quantifications, $n$ = 3–8 organoids per sample per genotype was used. In each organoid, 3–9 cortical structures with clear lamination patterns were examined. Data are represented in bar graphs as mean ± STD with individual data as dots; n.s. means $p > 0.05$, *$p < 0.05$, **$p < 0.01$, and ****$p < 0.0001$. Mixed-effects analysis multiple comparisons test with Tukey's multiple comparisons test as a post hoc was used for panel (C). Kruskal–Wallis ANOVA test with Dunn's multiple comparisons test as a post hoc was used for S6D Fig and S6E Fig. The numerical values that were used to generate the graphs can be found in S1 Data. BrdU, 5-bromo-2-deoxyuridine; EN, excitatory neuron; FOXP1, Forkhead Box P1; IPC, intermediate progenitor cell; KO, knockout; RGC, radial glial cell; ROI, region of interest; WT, wild type. (PDF)

**S7 Fig. Impaired late neurogenesis in 3-month (D100) organoids. (A)** Immunostaining images of D100 organoids showing expression of SOX2 for RGCs, SATB2 for ENs, and FOXP1 in a representative WT organoid. **(B)** Immunostaining showing IPCs (TBR2+) and ENs (SATB2+) at D100. **(C–F)** Immunostaining results showing percentage of RGCs, IPCs, and ENs at D100 normalized to all DAPI+ cells. For all quantifications, $n$ = 3–8 organoids per sample per genotype was used. In each organoid, 3–9 cortical structures with clear lamination patterns were examined. Data are represented in bar graphs as mean ± STD with individual data as dots; n.s. means $p > 0.05$, ***$p < 0.001$, and ****$p < 0.0001$ Kruskal–Wallis ANOVA test with Dunn's multiple comparisons test as a post hoc was used. Scale bar = 100 μM for S7A Fig. Scale bar = 300 μM for S7B Fig on the left and 100 μM for ROI selected images on the right. The numerical values that were used to generate the graphs can be found in S1 Data. EN, excitatory neuron; FOXP1, Forkhead Box P1; IPC, intermediate progenitor cell; RGC, radial

glial cell; ROI, region of interest; WT, wild type.
(PDF)

**S8 Fig. Neuronal morphology rescue by inhibiting expression of CTNND2. (A)** iNeuron formation from iPSCs. (**B**) Expression of CTNND2 is reduced upon CRISPRi transfection in 293T cells by RT-qPCR. **(C, D)** Successful neuronal differentiation at different time points (D5 and D14, respectively). Panel (**C**) shows ICC images of cells that were stained for early neuronal marker Tuj1 (TUBB3) at day 5, whereas panel (**D**) shows ICC images of cells that were stained for mature neuronal marker, MAP2, and cortical neuronal marker, CUX1 at day 14. (**E**) ICC images of GFP+ cells that are stained for CTNND2 and CUX1 at day 28. GFP+ cells in the WT and KO condition express empty GFP, whereas GFP+ cells in CTNND2 rescue conditions express dCas9 and gRNA targeting CTNND2. (**F**) Quantification of neuronal morphology in WT, KO, and CTNND2 CRISPRi conditions. $n$ = 4–5 per sample on glass coverslips were used, and $n$ = 10–14 per coverslip were quantified. Data are represented in bar graphs as mean ± STD with individual data as dots; $*p < 0.05$, and $****p < 0.0001$ Kruskal–Wallis ANOVA test with Dunn's multiple comparisons test as a post hoc was used. Scale bar = 100 μM for S8C–S8E Fig. The numerical values that were used to generate the graphs can be found in S1 Data. CRISPRi, CRISPR inhibition; ICC, immunocytochemistry; iNeuron, induced neuron; iPSC, induced pluripotent stem cell; KO, knockout; RT-qPCR, quantitative real-time PCR; WT, wild type.
(PDF)

**S9 Fig. Representative karyotype analysis for iPSCs G-banding technique showing normal chromosomes of metaphase cells across all three genotypes.**
(PDF)

**S1 Table. Sequencing read numbers, number of nuclei sequenced, number of expressed genes, and number of UMIs.**
(DOCX)

**S2 Table. List of DEGs per major cell type.**
(XLSX)

**S3 Table. List of NDD-related DEGs per major cell type.**
(XLSX)

**S4 Table. List of DEGs from individual clusters including those from bRGC subtypes.**
(XLSX)

**S5 Table. Gene ontology biological processes for bRGC subtypes.**
(XLSX)

**S1 Data. Excel Spreadsheet containing, in separate tabs, the underlying numerical data for Figs 1E, 3C, 3H, 3I, S1F, S5B, S5E, S6A, S6C, S6D, S6E, S6G, S7C, S7D, S7, S7F, S8B and S8F.**
(XLSX)

**S1 Raw Image.**
(PDF)

## Acknowledgments

We thank Dr. Peter Tsai and Konopka lab members for critical feedback on the manuscript. We thank Dr. Rolando Garcia for performing the karyotype analysis of the iPSC lines.

## Author Contributions

**Conceptualization:** Seon Hye E. Park, Genevieve Konopka.

**Formal analysis:** Seon Hye E. Park, Ashwinikumar Kulkarni.

**Funding acquisition:** Genevieve Konopka.

**Investigation:** Seon Hye E. Park.

**Project administration:** Genevieve Konopka.

**Supervision:** Genevieve Konopka.

**Validation:** Seon Hye E. Park.

**Visualization:** Seon Hye E. Park, Ashwinikumar Kulkarni.

**Writing – original draft:** Seon Hye E. Park.

**Writing – review & editing:** Ashwinikumar Kulkarni, Genevieve Konopka.

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
