## [Editor Report · Decision Letter 0]

26 Sep 2022

Dear Genevieve, 

I was glad to see the submission of your manuscript entitled "FOXP1 orchestrates neurogenesis in human cortical basal radial glial cells" for consideration at PLOS Biology.

I apologize that it took me a bit of time to get back to you. I was at the CSHL Neuronal Connectivity meeting a few weeks ago and was a bit backlogged upon my return. I've now had a chance to discuss your submission with my colleagues and with an academic editor with relevant expertise and I am writing to let you know that we would like to send your submission out for external peer review as a Short Report.

Before we can send your manuscript to reviewers, we will need you to complete your submission by providing the metadata that is required for full assessment. To this end, please login to Editorial Manager where you will find the paper in the 'Submissions Needing Revisions' folder on your homepage. Please click 'Revise Submission' from the Action Links and complete all additional questions in the submission questionnaire.

Once your full submission is complete, your paper will undergo a series of checks in preparation for peer review. After your manuscript has passed the checks it will be sent out for review. To provide the metadata for your submission, please Login to Editorial Manager (https://www.editorialmanager.com/pbiology) within two working days, i.e. by Sep 28 2022 11:59PM.

Feel free to email us at plosbiology@plos.org if you have any queries relating to your submission, or to reach out to me directly (kdickson@plos.org).

Kind regards,

Kris

Kris Dickson, Ph.D. (she/her)

Neurosciences Senior Editor/Section Manager

PLOS Biology

kdickson@plos.org

---

## [Decision Letter · Decision Letter 1]

4 Nov 2022

Dear Dr Konopka,

Thank you for your patience while your manuscript "FOXP1 orchestrates neurogenesis in human cortical basal radial glial cells" was peer-reviewed at PLOS Biology. Your manuscript has been evaluated by the PLOS Biology editors, an Academic Editor with relevant expertise, and by two independent reviewers.

As you will see in the reviewer reports, which can be found at the end of this email, although the reviewers find the work potentially interesting, they have also raised a substantial number of important concerns. Based on their specific comments and following discussion with the Academic Editor, it is clear that a substantial amount of work would be required to meet the criteria for publication in PLOS Biology. While recognizing that the reviewers requests mean that a substantial amount of work is still required, given our and the reviewer interest in your study, we would be open to inviting a comprehensive revision of the study that thoroughly addresses all the reviewers' comments. In particular, the key issue seems to come down to how you are interpreting FOXP1 expression and the cell types of interest, and we'd ask that you consider the reviewers' thoughts on how you are currently interpreting the work. Our Academic Editor also felt that further comprehensive immunostainings and additional investigations of the aRGCs and snRNA-seq data are warranted and could address much of the reviewer technical concerns. 

Given the extent of revision that would be needed, we cannot make a decision about publication until we have seen the revised manuscript and your response to the reviewers' comments. Your revised manuscript would need to be seen by the reviewers again, but please note that we would not engage them unless their main concerns have been addressed. 

We appreciate that these requests potentially represent a great deal of extra work, and we are willing to relax our standard revision time to allow you 6 months to revise your study. Please email us (plosbiology@plos.org) if you have any questions or concerns, or envision needing a (short) extension.

**IMPORTANT - SUBMITTING YOUR REVISION**

*Resubmission Checklist*

*Published Peer Review*

*PLOS Data Policy*

*Blot and Gel Data Policy*

Thank you again for your submission to our journal. I apologize that this wasn't the fasted of reviews and hope that our editorial process has been constructive thus far. I welcome your feedback at any time. Please don't hesitate to contact me if you have any questions or comments.

Sincerely,

Kris

Kris Dickson, Ph.D., (she/her)

Neurosciences Senior Editor/Section Manager

PLOS Biology

kdickson@plos.org

REVIEWS:

Reviewer's Responses to Questions

Do you want your identity to be public for this peer review?

Reviewer #1: No

Reviewer #2: No

Reviewer #1: FOXP1 orchestrates neurogenesis in human cortical basal radial glial cells

Park et al. present a study aimed at understanding the human-specific role of FOXP1 in cortical development and linking its loss of function to human neurodevelopmental disorders. The study is done in cortical organoids, which is advantageous in many aspects to rodent models used in past studies of this gene. The study relies on snRNA data images to depict the gene's role in the developmental trajectory of cortical progenitors, focusing on a subclass of progenitors that occupy the main proliferative zone in the human cortex. The study is of great interest, but it suffers from some problems, particularly the nomenclature of the human progenitor subclasses and the experiments that reveal decreased proliferation and differentiation of bRGCs following FOXP1 loss. 

Specific remarks: 

NDD- Neurodevelopmental disorders, the acronym is used without its meaning.

Figure 1: Based on the strategy to produce KO2, one would expect to detect GFP expression in the sections. Is it there? (looking at Figure 3B, no GFP is expressed in KO2?). If GFP is expressed in KO2, wouldn't it be a great tool to look at the location proliferation and progeny of the mutated cells or a way to isolate the mutant cells? Why is this option not being used throughout the study?

Figure S1/Figure 2: Based on others, D25 oRG's are rare at the developmental time, yet FOXP1 expression is abundant in the VZ. Likewise, at D40 (Figure 1D), the VZ is the prominent proliferative region, and only a few FoxP1 cells are seen in more basal locations. Previous studies show HOPX and FOXP1 colocalize to cells in the osVZ in human sections. Given this, why do the authors consider FOXP1 to be a bRG-enriched gene in their organoids? With the small proportion of bRG's at D40 when most of the study was conducted and the high expression in cells that are not bRG, I find the statement the study focuses on bRG a little problematic.

Figure 2. In the beautiful layered images of D40 (Figure S1C) it seems like the developmental trajectory is FoxP1+(aRG?)-> TBR2+(IPC)->CTIP2+(EN). The scRNA data shows a pseudo time in with aRG->bRG->IPC->EN. Is FOXP1 better defined, at this developmental stage, as a typical aRG rather than a bRG gene?

The confusion is partly due to the nomenclature used (more suitable for rodent brains rather than human brains). Based on some papers that the authors cite (Kanton et al., 2019; Nowakowski et al., 2017), the early cortical development trajectory is RG->IPC->EN, and in later stages, RG-> oRG->IPC (both oRG and IPC have the neurogenic ability). Later, RG will give rise to tRG, which will contribute to a noncontinuous scaffold with the basal processes of the oRG cells. In this study, the authors state that they aim at the beginning of neurogenesis. Yet, the cell line they are focusing on is highly neurogenic at later stages, and the trajectory depicted is aRG->bRG->IPC, consistent with late neurogenesis. 

Figure S2: "We observed FOXP1 expression in 79.35% of the bRGCs (Figure S2C)". In figure S2, which are the bRGs? The figure shows RG, IPC, and EN. Can a list of genes defining clusters 17 and 25 (Figure 2) be presented in the main figure? 

Figure 3: 

The study by Pearson et al 

(https://www.ncbi.nlm.nih.gov/pmc/articles/PMC8397815/ ) showed that Foxp1 promotes aRG maintenance in the mouse cortex by promoting vertical, symmetric self-renewing cell divisions. I expected the study of FOXP1 on the proliferation and progeny of aRG and bRG to be conducted with extra care. Indeed, the strategy to use viruses for sparse labeling is very suitable for the study. Yet, the identification of bRG is based on morphology and negative criteria of low expression of Tbr2. The morphology study is done on 16 �M thick slices, which may be why many processes are not continuous (Figure 3B). Given the 3D nature of the organoid VZ, I believe that a thick slice will be more reliable.

Furthermore, morphology is problematic in later stages; the authors show that in the FOXP1 KO, adhesion molecules are down-regulated. Would that potentially cause changes in possesses morphology? The second part of the figure is even more problematic. Staining with HOPX reveals the low abundance of bRGs at the time point but also raises the question of why this was not used as a positive criterion in Figure 3A to identify the bRG that are supposed to be the center of the study. The progeny study is based on location and is not reliable, in my opinion. Given that a viral infection is so elegantly done, clonal analysis of the sparsely infected cell is more suitable. 

Reviewer #2: Park et al., present their work investigating the role of FOXP1 in basal radial glial cells during human corticogenesis, using cortical organoids derived from FOXP1-/- iPSCs. This work is novel and significant as it begins to define the role of FOXP1 in human cortical progenitors, and performs scRNA Seq to assess the impact of loss of FOXP1 in human cortical cell populations. A particularly interesting observation is that there are 2 identified populations of FOXP1+ bRGCs and that FOXP1 may regulate different pathways in each population. Also, that bRGC may be especially vulnerable to ASD. However, I have issues with the authors interpretation of their results that need to be addressed. 

The statistical analyses and supplemental data are sufficient as are the details in the materials and methods. 

Minor concerns/edits:

* Authors mix up descriptions of direct and indirect neurogenesis in the 1st paragraph of the introduction. 

* The experiments are performed using day 40 organoids, however there appear to be very few bRGCs present at this time point - at least as shown by the immunohistochemical analyses in Fig 1 and S1. The image quality in general needs to be improved as it is difficult to see some of the stains at the current resolution. 

* The authors should include improved images of FOXP1+ bRGCs in their organoids. 

* Figure 2 - the authors use expression of SOX2+PAX6+HES5+ transcripts as 0 in their pseudotemporal analysis to represent aRGCs. However, these trancripts should also be present in bRGCs. I found the descriptions of Figure 2 difficult to follow.

* Figure 2E the authors say "the bRGCs showed reduced gene expression changes from root cells with the loss of FOXP1' however the density plots suggest a shift towards 0 for bRGCs. 

* The authors conclude that Figure 2 suggests "impaired differentiation" of bRGCs with loss of FOXP1. This is vague - how is it impairing differentiation? The authors report that certain bRGC genes are decreased - does this suggest the bRGC have differentiated precociously or are more "stem-like"? 

* Figure 3 - using low Tbr2 as a marker of bRGCs is subjective. Why not use Pax6 or Hopx? 

* At the magnification used in some of these images it is difficult to distinguish whether the bRGCs are outside of the ventricular zone. Especially as the images are fuzzy. 

* The images in 2E are difficult to interpret due to low quality. The images in 2G should be in color to show merged image. It is not clear that the HOPX+ cells analyzed are outside of the ventricular zone (where bRGCs should reside). 

* Figure S5 - the PTPRZ staining does not look specific to bRGCs. 

* Figure 3 - the authors should quantify the % of BrdU+ HOPX+ cells over HOPX+ cells rather than DAPI+ to give a more accurate reflection of the proliferating bRGCs. Also, the authors appear to imply that a BrdU+ bRGC is self-renewing, however this assay cannot be used to determine self-renewing capacity, only that the cell is proliferative. 

* Figure 3 would be improved by an analysis of the total numbers of aRGCs, bRGCs, INs and neurons in wt and FOXP1-/- organoids. 

* A more thorough analysis of the other neuronal populations is required to make a statement regarding impact of FOXP1 loss on neurogenesis. 

* It seems a shame that the data in Figure S8 C is not in a main figure. 

Major issues:

* The analysis in Figure 3 of adjacent cells relative to HOPX+ cells cannot be used to determine indirect or direct neurogenic divisions. To infer whether bRGCs have undergone a direct or indirect neurogenic division, clonal analyses need to be conducted using time lapse imaging with sparsely labelled GFP+ positive cells. The proximity of one cell to another does not indicate that one cell gave rise to the other. 

* The major issue I have is the failure to consider that FOXP1+ bRGCs are derived from FOXP1+ aRGCs which will also be impacted by the mutation. The authors cannot distinguish between whether the changes they are seeing in the bRGC population are due to loss of FOXP1 in the bRGCs or its loss in the aRGC population that they arose from. For instance, the increased IPCs reported in FOXP1-/- organoids could be as a result of early generation from the aRGC population. Numbers of aRGCs in the absence of FOXP1 are not reported. Similarly, the neuronal population the authors analyze also express FOXP1. Therefore, the impact on this population reported could be due to changes in the progenitor population (both aRGCs and bRGCs) or the neurons themselves. 

* A more thorough analysis of cell populations in the absence of FOXP1 is required at different time points. Images should be improved, and interpretations should reflect the loss of FOXP1 in all cell types. 

* I recognize that with an unconditional knock out it is not possible to determine the bRGC specific role of FOXP1. However, I do believe that there is value in reporting the loss of function phenotypes observed in human cortical organoids. These analyses can be improved and the conclusions re-interpreted to reflect the contribution of FOXP1+ aRGCs and neurons.

---

## [Decision Letter · Decision Letter 2]

2 Jun 2023

Dear Dr Konopka,

Thank you for your patience while we considered your revised manuscript "FOXP1 orchestrates neurogenesis in human cortical basal radial glial cells" for publication as a Short Report at PLOS Biology. This revised version of your manuscript has been evaluated by the PLOS Biology editors, by the original reviewer 2, and by the Academic Editor. 

As you will see below, Reviewer 2 is fully satisfied by the revision and suggest we accept the manuscript. Our Academic Editor is also satisfied by the changes made in the revision and in response to reviewer 1 and we are therefore likely to accept this study. However, before we can editorially accept your manuscript, we need you to address a few remaining data and other policy-related requests, which I outline below, in another revision that we think will not take very long. 

**Please address the following editorial requests: 

1) ETHICS STATEMENT: thank you for providing an ethics statement in your methods section. Can you please update this to include the approval number of the protocol that was approved by the UT SCRO?

2) DATA AVAILABILITY: Thank you for providing your RNA-seq data on the GEO repository. In addition to this, you may be aware of the PLOS Data Policy, requires that all data be made available without restriction: http://journals.plos.org/plosbiology/s/data-availability. For more information, please also see this editorial: http://dx.doi.org/10.1371/journal.pbio.1001797 (we need you to also provide the underlying data presented in your figures that are not related to the RNA seq data already provided)

a - Supplementary files (e.g., excel). Please ensure that all data files are uploaded as 'Supporting Information' and are invariably referred to (in the manuscript, figure legends, and the Description field when uploading your files) using the following format verbatim: S1 Data, S2 Data, etc. Multiple panels of a single or even several figures can be included as multiple sheets in one excel file that is saved using exactly the following convention: S1_Data.xlsx (using an underscore).

b Deposition in a publicly available repository. Please also provide the accession code or a reviewer link so that we may view your data before publication. 

>>Regardless of the method selected, please ensure that you provide the individual numerical values that underlie the summary data displayed in the following figure panels as they are essential for readers to assess your analysis and to reproduce it:

FIg 1E; Fig 3C,H;

Fig S1F; Fig S5B,E; Fig S6C-E,G; Fig S7C-F; Fig S8B,F;

>>Please also ensure that figure legends in your manuscript include information on where the underlying data can be found, and ensure your supplemental data file/s has a legend.

>>Please ensure that your Data Statement in the submission system accurately describes where your data can be found.

3) BLOT AND GEL REPORTING: Thank you for providing, as a supplemental file, the raw images related to the western blots presented in your study. It looks as though these images have been slightly cropped. If contained in the original image, please adjust these images to include the complete blot (some of the edges seem cut off)

4) DATA NOT SHOWN: Please note that per journal policy, we do not allow the mention of "data not shown", "personal communication", "manuscript in preparation" or other references to data that is not publicly available or contained within this manuscript. Please either remove mention of these data or provide figures presenting the results and the data underlying the figure(s).

>> I noticed one instance of this, on page (SATB2 and vGLUT2). 

We expect to receive your revised manuscript within two weeks. 

*Published Peer Review History*

*Press*

Sincerely,

Luke

Lucas Smith, Ph.D.

Senior Editor,

lsmith@plos.org,

PLOS Biology

Reviewer remarks:

Reviewer #2: The authors have appropriately addressed my questions and concerns, and therefore I think this manuscript should be accepted.

---

## [Editor Report · Decision Letter 3]

21 Jun 2023

Dear Dr Konopka,

Thank you for the submission of your revised Short Report "FOXP1 orchestrates neurogenesis in human cortical basal radial glial cells" for publication in PLOS Biology and thank you for addressing our last editorial requests. On behalf of my colleagues and the Academic Editor, Madeline Lancaster, I am pleased to say that we can in principle accept your manuscript for publication, provided you address any remaining formatting and reporting issues. These will be detailed in an email you should receive within 2-3 business days from our colleagues in the journal operations team; no action is required from you until then. Please note that we will not be able to formally accept your manuscript and schedule it for publication until you have completed any requested changes.

**As a quick note - I see that you have provided the underlying data for your figures, and referenced this in the figure legends (S1_data) - Thank you! I noticed you also reference this data file throughout the manuscript text. That is totally fine - but if you think it is cumbersome, it would also be OK if you remove the references to the underlying data in the text. We will leave it up to you to change this or not. (If you do change this, please leave the references to S1_data in your figure legends). 

PRESS

Sincerely, 

Lucas Smith, Ph.D.

Senior Editor

PLOS Biology

lsmith@plos.org